# Transport Reagents through the Pore Structure of a Membrane Catalyst under Isothermal and Non-Isothermal Conditions

**DOI:** 10.3390/membranes11070497

**Published:** 2021-06-30

**Authors:** Natalia Gavrilova, Sergey Gubin, Maria Myachina, Valery Skudin

**Affiliations:** 1Department of Colloid Chemistry, Faculty of Natural Science, D. Mendeleev University of Chemical Technology of Russia, Miusskaya Sq. 9, 125047 Moscow, Russia; mmyachina@muctr.ru; 2Department of Chemical Technology of Carbon Materials, Faculty of Petroleum Chemistry and Polymers, D. Mendeleev University of Chemical Technology of Russia, Miusskaya Sq. 9, 125047 Moscow, Russia; liod.sniper@gmail.com (S.G.); skudin@muctr.ru (V.S.)

**Keywords:** membrane catalyst, diffusion transport mode, membrane reactor, molybdenum carbide, conventional reactor, dry reforming of methane, non-isothermal Knudsen diffusion, diffusion coefficient, thermal creep

## Abstract

The article presents the results of an experimental comparison of methane transport in the pore structure of a membrane catalyst under isothermal and non-isothermal Knudsen diffusion conditions. It is shown that under the conditions of non-isothermal Knudsen diffusion in the pore structure of the membrane catalyst, there is a coupling of dry reforming of the methane (DRM) and gas transport, which leads to the intensification of this process. The reasons for the intensification are changes in the mechanism of gas transport, an increase in the rate of mass transfer, and changes in the mechanism of some stages of the DRM. The specific rate constant of the methane dissociation reaction on a membrane catalyst turned out to be an order of magnitude (40 times) higher than this value on a traditional (powder) catalyst.

## 1. Introduction

The advantage that distinguishes the membrane catalyst from the traditional one is the ability to control the transport of gases and their mixtures during the implementation of heterogeneous catalytic processes. The thermodynamic analysis of transport in membranes, published more than 30 years ago in the monograph [1], showed that the coupling of a chemical reaction with transport in pores can be accompanied by a significant change in the kinetic parameters of processes in membrane reactors. However, lack of systematic kinetic studies of catalytic reactions in the presence of membranes limits the validation of this prediction. The basic structure of a membrane catalytic reactor, in contrast to a traditional reactor, is that it allows a given catalytic reaction to be carried out in several modes by controlling the supply of reagents and the removal of the reaction mixture from it.

Incorporating a membrane in a reactor always divides its reaction volume into two parts, creating new possibilities for controlling a process. The catalyst can be deposited in any of these parts or, directly, on the membrane. The modes of operating membrane catalytic reactors can be divided into three. When both reactants and products are respectively fed into the reactor and collected from the reactor in the form of a mixture, then this is referred to as a contactor mode. If two or more reactants are fed separately into the membrane reactor, but products of the reaction are collected as a mixture or separate streams, then this is referred to as distributor mode. When the reactants are fed into the membrane reactor as a mixture, and the reaction products or one of them is removed separately from the stream of the unreacted mixture, then this is referred to as extractor mode. These varying modes of operation are rather arbitrary. For example, for gas–gas reactions, the distributor mode in membrane reactor can be considered as one of the contactor mode’s subdivisions. In a reactor with a membrane catalyst, several types of contactor modes can be achieved [1,2,3]. Contactor mode cannot be realized in a reactor with a selective membrane and fixed bed of a conventional catalytic reactor. Although the contactor mode is promising, it is not widely studied. Extractor and distributor modes can be exploited in reactors with membrane catalyst and in reactors with selective membrane and traditional catalyst.

Extractor mode is the most widely studied and is not needed in the presentation. The advantage of the distributor mode is the prevention of the reactants’ escape “without participating in the targeted reaction” [2]. Another concept is characterized by the forced supply of premixed reagents and their transport through the pore structure of a membrane catalyst [4]. Such a mode is especially promising for fast reactions, in which kinetic limitations are often manifested. Detailed experimental evidence of the above can be found elsewhere [5,6,7]. However, the published materials are limited to a qualitative assessment of the improvement in the contact of reactants with the catalyst surface, without a deep explanation of the reasons for the observed phenomena.

In Ref [8], the results of a kinetic study of the contactor mode with a forced transport of reagents in a reactor with a membrane catalyst were presented. Comparison of the results of this experiment with the results obtained on a traditional catalyst of the same chemical composition made it possible to quantitatively confirm the effect of intensification in the dry reforming of methane (DRM). The authors proposed a hypothesis explaining that non-isothermal Knudsen diffusion is responsible for this intensification [9]. In the presence of a temperature gradient on the surface walls of channels in the pore structure of a membrane (diaphragm) separating two containers filled with a gas medium, a phenomenon called thermal slip (thermal transpiration or thermal creep) is observed. In this case, the method of creating a temperature gradient does not affect the essence of the process, and its intensity is determined by the Knudsen number and can vary over a wide range of its values.

In this work, in order to compare diffusion under isothermal and non-isothermal conditions, the contactor mode in a reactor with a membrane catalyst was modified. The premixed reagents were passed not through the pore structure of the membrane catalyst, but over its outer surface, and were removed from the shell side of the reactor. It was assumed that the transport characteristics of the reagents of DRM in the channels of the pore structure of membrane and traditional catalysts will be similar. To determine the effective diffusion coefficient under isothermal conditions, the experiment was carried out at a constant temperature (200 °C), under which the reaction is thermodynamically impossible. Non-isothermal diffusion experiments were carried out at the favorable temperature range for DRM reaction (from 800 °C to 890 °C).

The diffusion of methane under isothermal conditions was studied using its mixtures with nitrogen and carbon dioxide. Diffusion of carbon dioxide was investigated in a mixture with nitrogen. These experiments were performed at a constant temperature (200 °C). During the experiments, the methane flux density was determined (the diaphragm method [9]) through the pore structure of the membrane catalyst.

Under non-isothermal conditions, the methane flux in the pores of the membrane catalyst was determined from the results of a kinetic experiment in the DRM, as the amount of methane that was reacted. Dry reforming of methane is an endothermic reaction; it occurs at a high temperature and is accompanied by a number of side reactions [10,11,12]. In this process, methane participates in only one reaction—dissociation into carbon and hydrogen, which occurs on the surface of the pores of the catalytic layer, which makes it possible to consider this amount to be corresponding to the amount of diffused substance.

The aim of this work was to experimentally determine and compare the transport characteristics on a membrane catalyst under isothermal and non-isothermal conditions (DRM conditions).

## 2. Materials and Methods

### 2.1. Synthesis of Catalyst

Powder catalyst Mo_2_C was prepared by temperature decomposition of molybdenum blue xerogels in inert atmosphere at 900 °C.

A Mo_2_C/Al_2_O_3_ membrane catalyst was prepared in two steps: first, a layer of the precursor tungsten oxide was formed on the substrate (ceramic microfiltration membrane) by chemical vapor deposition (CVD), which was then converted into molybdenum carbide using temperature-programmed carburization (TPC).

Tubular microfiltration membranes made of α-Al_2_O_3_ with a length of 20 cm, an outer diameter of 10 mm, and a wall thickness on the order of approximately 1 mm were used as the substrate. The layer was deposited onto the outer side of the microfiltration membrane. The deposition of molybdenum oxides was performed in a “cold-wall” CVD reactor. Molybdenum hexacarbonyl Mo(CO)_6_ was used as the starting volatile molybdenum compound. The deposition was carried out in an inert atmosphere (nitrogen of the special purity grade). The process temperature (250 °C) is the temperature at which deposition occurs on the outer surface of the substrate.

### 2.2. Catalytic Activity of Membrane and Conventional Catalysts Determination

Catalytic studies were carried out in reactors with a membrane catalyst in the mode of a contactor with diffusion transport of reagents and in a traditional reactor with a fixed catalyst bed.

A sample of a membrane catalyst with a long outer working surface of 5.5 cm and a diameter of 1 cm contained 0.1535 g of Mo_2_C. One end of the membrane catalyst was sealed with a gas-tight material, and the other was hermetically fixed on a special holder. Thus, the membrane catalyst divided the reaction space inside the reactor into two parts (Figure 1).

The temperature inside the reactor was recorded using two thermocouples located on both sides of the membrane catalyst. In the course of kinetic studies, the junctions of both thermocouples were located at the same level corresponding to the average temperature values in the tube side of the membrane catalyst. The readings of the thermocouple located in the tube side were taken as the process temperature. 

A mixture of CH_4_ and CO_2_ (with a molar ratio of 1:1) was fed into the shell side of the membrane reactor. The transport of reagents through the pore structure of the membrane catalyst was carried out by diffusion. The reaction mixture was removed from the reactor shell side as retentate. 

The flow rate of the initial mixture was controlled using two EL-Flow mass flowmeter “Bronkhorst High-Tech B.V.” (“Bronkhorst High Tech”, AK Ruurlo, Netherlands). The composition of the products (CH_4_, CO_2_, CO, and H_2_) at the outlet of the reactor was analyzed using a Crystallux-4000M gas chromatograph (CJSC SKB “Chromatec”, Yoshkar-Ola, Russia). The flow rate of the mixture at the reactor outlet was measured using an ADM—2000 flow meter (Agilent, Santa Clara, CA, USA). The amount of water vapor generated in the dry reforming of methane was determined from the mass balance according to experimental data.

Experiments in the membrane reactor were carried out at the temperature range of 820–890 °C. In this case, the flow rate of the reagent mixture was changed from 50 mL/min to 150 mL/min. The powdered (traditional) catalyst was tested at the temperature range of 830–900 °C in a quartz reactor with a thermocouple jacket located along the axis of the reactor. The powdered catalyst was placed on a mesh shelf in the annular space of the reactor. 

The catalyst was premixed with crushed quartz with a maximum particle size of less than 200 μm at a volume ratio of 1:1. A mixture of quartz with a powdery catalyst was placed on the shelf in the medium temperature zone for the catalyst layer above the layer of coarsely ground quartz. A layer of crushed quartz was also placed above the catalytic layer. The temperature in the reactor was monitored by a thermocouple placed in a thermocouple jacket of the reactor. The mass of the Mo_2_C powder during these experiments was 0.261 g. The flow rate of the reagent mixture was varied from 35 to 162 mL/min.

The activity of the membrane and traditional catalysts was estimated by the value of the rate constant (Equation (1)) for the rate-limiting stage of DRM process—methane dissociation reaction:(1)$$k_{m}=\frac{m_{a.c.}}{Q}$$

$k_{m}$—specific rate constant, cm^3^ [CH_4_(n.c.)]/[g (a.c.)*s]; *m*_a.c._—weight of catalyst in the traditional and membrane catalysts in kinetic experiment, g; *Q*—volumetric flow rate of methane reacted (2) under normal conditions, cm^3^ CH_4_(n.c.)/s.
(2)$$Q=V_{{\mathrm{CH}}_{4}}^{0}-V_{{\mathrm{CH}}_{4}}^{1}$$

$V_{{\mathrm{CH}}_{4}}^{0}$, $V_{{\mathrm{CH}}_{4}}^{1}$—methane flow rate in cm^3^ CH_4_(n.c.)/s at the reactor inlet and outlet, respectively.

The reciprocal of the rate constant corresponding to the conditional contact time was used as the main variable in the graphical presentation of kinetic data. 

Reacted methane flux density ($J_{{\mathrm{CH}}_{4}}^{react}$), was determined using the material balance of DRM according to the Equation (3):(3)$$J_{{\mathrm{CH}}_{4}}^{react}=\frac{Q}{F_{work}}=\frac{Q}{\pi d_{in}\times l_{work}}$$

$F_{work}$—outer working surface of a membrane catalyst in cm^2^; $d_{in}$—outer diameter of membrane catalyst in cm; $l_{work}$—the length of the membrane catalyst involved in the reaction in cm. For the traditional catalyst $F_{work}$ corresponds to the cross-section of the annular space of the traditional reactor.

### 2.3. Experimental Determination of the Effective Diffusion Coefficient under Isothermal Conditions (Diaphragm Method)

One of the methods for determining the effective diffusion coefficient is the so-called diaphragm method, which was developed by Reuter and coworkers [9,13] and adapted in this work for a membrane reactor.

The essence of this method is that the membrane catalyst is used as a diaphragm and is installed hermetically in the reactor, dividing reaction space into two parts: shell side and tube side. Each of these parts has its own inlet and outlet. The test gas mixed with an inert carrier gas is fed into one of the parts (chambers) of this cell, and only the inert carrier gas is fed into the other part. At the same pressure in both chambers, the exchange of substances between the chambers can occur only due to the diffusion of gases through the diaphragm (membrane).

Figure 2 shows a schematic diagram of a membrane reactor adapted to determine the effective diffusion coefficient.

The transport of methane and carbon dioxide through the pore structure of the membrane catalyst was studied. Nitrogen was supplied to one of the parts (chambers) of the internal space of the reactor, formed during the installation of the membrane catalyst (diaphragm), and a mixture of methane or carbon dioxide with equal nitrogen concentration was supplied to the other.

The composition of streams entering and leaving each chamber was determined using a “CrystalLux 4000M” chromatograph. The volumetric flow rate of each of the streams was also measured. The measurement results were used to calculate the molar flux of the target mixture component.
(4)$$b=Q^{in}c^{in}-Q^{out}\;c^{out}$$

The flux density and effective diffusion coefficient were calculated using the equations:(5)$$J_{i}=\frac{b}{F_{work}}$$
(6)$$D^{*}=\frac{bd}{\left(c^{\mathrm{in}}-c^{\mathrm{out}}\right)F_{\mathrm{work}}}$$

*b*—the amount of substance diffused through the working surface of the membrane catalyst per unit time, mol/s; $J_{i}$—flux density, mol/(s·cm^2^); *d*—thickness of the catalytic layer of the membrane catalyst, cm; $Q^{in}$, $Q^{out}$—the flow of the mixture at the inlet and outlet from the cell, respectively, cm^3^/s; $c^{in}$ and $c^{out}$—concentrations of the reagent at the inlet and outlet streams in the chambers, mol/cm^3^; $F_{work}$—outer working surface area of tube type diaphragm (membrane catalyst), cm^2^.

## 3. Results

### 3.1. Catalyst Characterization

In this work, two catalyst forms were compared: powder and membrane. Powder catalyst (hereinafter referred to as the traditional catalyst (TC)) consisted of particles containing carbon and molybdenum in the ratio C:Mo = 1:1, forming phases β-Mo_2_C and η-Mo_2_C. This catalyst was obtained via sol-gel method using molybdenum blue as a precursor. The solid phase of these dispersions was isolated by removing water during drying, and then calcined at a temperature of 900 °C in an inert atmosphere for 1 h.

The membrane catalyst (MC) was prepared using chemical vapor deposition (CVD) method, in a reactor with “cold” walls at a substrate temperature of 250 °C, by precipitating of molybdenum dioxide (MoO_2_) from a mixture of molybdenum hexacarbonyl vapors in nitrogen onto corundum (α-Al_2_O_3_) microfiltration membrane. By treating the membrane in a flow in a mixture of hydrogen and methane through the process of temperature-programmed carburization, molybdenum dioxide was reduced to molybdenum carbides. The main characteristics of catalysts are presented in Table 1.

The deposition of molybdenum dioxide by CVD method was carried out in a mode that ensures the formation of a massive layer on the outer surface of a tubular microfiltration membrane. Formation of a massive layer was needed to ensure the maximum approximation of its properties and to compare them to that of the active component of the traditional catalyst.

The cross-sectional micrograph (Figure 3a) shows the structure of the membrane catalyst. Large particles with a size of more than 10 μm (in the right quarter of the image) correspond to the main part of the ceramic membrane, above which (in the central part) the outer (selective) layer is clearly visible. The lighter areas in the photomicrograph correspond to the molybdenum carbide layer. It can be seen that the carbide layer is separated by a clear boundary. One part of it (relatively small) is distributed over the selective layer of the microfiltration membrane, and the other is a bulk carbide. The bulk of the catalytically active substance is contained, precisely, in this layer, which has a thickness of about 10–14 μm. Despite the fact that large pores are visible in the image of the outer surface, the molybdenum carbide layer is a continuous coating.

Additional information on the preparation of membrane catalysts can be found in [14,15]. Thus, the active substance in both catalysts has the same chemical and phase compositions.

### 3.2. Dry Reforming of Methane in Membrane and Traditional Reactors

The mode of transport of reagents in a reactor with a membrane catalyst is characterized by how the reagents are fed into the reactor (in the form of a mixture or separate flows) and how the reaction products are removed from it as well as what driving force controls the transport of reagents to the catalyst surface.

In the diffusion transport mode in a membrane catalyst reactor, reagents and products are fed and removed in the form of mixture. Moreover, under the conditions of this experiment, the mixture of reagents passes through the shell side of the reactor over the outer surface of the tubular MC, on which the active component, molybdenum carbide, is deposited. Reaction mixture leaves the reactor (without separating or removing individual components from it by means of a membrane). This mode is demonstrated in Figure 1.

The catalytic active component of MC is placed on the outer membrane surface in the form of a massive layer under conditions of relatively low penetration of $Mo_{2}C$ into the pore space. In addition, in all experiments in the membrane reactor, the permeate line, which is connected to the tube side (interior volume of the tubular membrane catalyst), was closed. Formally, the transport of reagents into the catalytic pore of the membrane catalyst should be identical to the transport in a traditional catalyst. In both cases, the delivery of reagents to the catalyst surface can be carried out by diffusion, the driving force of which is the concentration gradient. It was assumed that the results of the kinetic experiment in this mode will be similar on both catalysts (TC and MC). On the other hand, this would explain the advantage of the forced transport mode in a reactor with a membrane catalyst, which we published earlier [15].

### 3.3. Main Stages of Dry Reforming on Traditional and Membrane Catalysts

The DRM reaction (Equation (7)) can be considered as a parallel-sequential set of a number of stages [10,16]:(7)$${\mathrm{CH}}_{4}+{\mathrm{CO}}_{2}↔2\mathrm{CO}+2H_{2},\;\Delta H_{298}^{0}=247\;\mathrm{kJ}/\mathrm{mol}\;(I)$$

The reaction of dissociation of methane (Equation (8)) is considered to be the limiting stage in both steam and dry reforming of methane:(8)$${\mathrm{CH}}_{4}↔C+2H_{2},\;\Delta H_{298}^{0}=75\mathrm{kJ}/\mathrm{mol}\;(\mathrm{II})$$

Dissociation of methane (stage II) proceeds in the presence of a catalyst. Obviously, carbon under the temperature conditions of this experiment can exist only in solid form, and therefore it is localized on the catalytic surface. The hydrogen passes into the gas phase of shell side.

Reaction (Equation (9)) of hydrogen and carbon dioxide is non-catalytic and homogeneous and proceeds in the gas phase at a high rate, almost always reaching equilibrium concentration values:(9)$${\mathrm{CO}}_{2}+H_{2}↔\mathrm{CO}+H_{2}O,\;\Delta H_{298}^{0}=-41\;\mathrm{kJ}/\mathrm{mol}\;(\mathrm{III})$$

As a result of reaction (Equation (9)), carbon monoxide and water vapor are formed. It can be assumed that water vapor and carbon dioxide can participate in reactions with carbon deposits on the surface of the catalyst pores (Equations (10) and (11), respectively). When carbon dioxide and water vapor interact with carbon, the products of DRM are formed.
(10)$$C_{s}+H_{2}O↔\mathrm{CO}+H_{2},\Delta H_{298}^{0}=131.3\;\mathrm{kJ}/\mathrm{mol}\;(\mathrm{IV})$$
(11)$$C_{s}+{\mathrm{CO}}_{2}↔2\;\mathrm{CO},\;\Delta H_{298}^{0}=172.4\;\mathrm{kJ}/\mathrm{mol}\;(V)$$

These reactions are heterogeneous, and carbon was formed on the active catalytic sites. The interaction of carbon dioxide with carbon deposits produces only carbon monoxide, while water vapor with carbon produces both target products (CO and H_2_). However, when analyzing the DRM process in publications, there is more often an assumption about the interaction of carbon dioxide with carbon deposited on the catalytic sites through reaction (IV).

The total expression of all reactions at the intermediate stages of the DRM should correspond to the reaction I. Note that in reaction (IV) carbon (C) and water vapor (H_2_O) interact, which in the process of DRM can be considered to be intermediate products. If the process is incomplete, these intermediate products can become undesirable and detrimental to the catalyst.

### 3.4. Kinetic Experiments

Figure 4 shows that the conversion of methane on the traditional and membrane catalysts differs significantly under the same contact time. This is indicative of DRM intensification, which was observed in [7] on traditional and membrane catalysts with an active component in the form of tungsten carbide (WC). This phenomenon was reproduced on an active substance in the form of molybdenum carbide (Mo_2_C). Namely, the contact time required to achieve the same methane conversion on a membrane catalyst turned out to be an order of magnitude shorter than the contact time on a conventional catalyst.

The apparent activation energy values $E_{app}$ for methane in membrane and conventional reactors also (Table 2) differ markedly. In a reactor with a membrane catalyst, the activation energy for reaction (II) is half that in a reactor with a traditional catalyst. This decrease in the apparent activation energy ($E_{app}$) can be interpreted as a result of the occurrence of a chemical reaction on a membrane catalyst within a diffusion region [12,17]. However, the rate constant of reaction (II) proceeding in the pore space of the catalytic layer of MC is an order of magnitude higher than the rate constant in the traditional (powder) catalyst (40 times).

If the traditional analysis of the efficiency of heterogeneous catalysts was applied to the results of a kinetic experiment, it can be concluded.

To assess the efficiency of a heterogeneous catalyst in the diffusion region of the DRM process, one can use the equation
(12)$$\eta =\frac{1}{L}\sqrt{\frac{RD_{eff}}{2k}}$$
where: *η*—the efficiency, defined as the ratio of the specific rate constant of reaction (II) under diffusion limitation conditions (on a membrane catalyst) to the rate constant in the same reaction on the same catalyst in the kinetic region (on a traditional, powdery catalyst); *L*—the pore length; $D_{eff}$—effective diffusion coefficient in the pore; *k*—specific rate constant of reaction (II) [18].

It follows from the equation that the catalyst efficiency with an increase in the specific rate constant by an order of magnitude can remain within reasonable values only with a simultaneous increase in $D_{eff}$. In other words, the transport of the components of the reaction medium in the pores of the membrane catalyst should occur according to a mechanism that provides a more intensified mass transfer than on a traditional catalyst. 

The trendlines shown in Figure 5 characterize the dependences according to which the concentrations of reagents and reaction products change as methane is consumed on both catalysts. These patterns (trendlines) were obtained by approximating the experimental points on traditional and membrane catalysts with a wide variation in temperatures and flow rates of the reagent mixture. The location of experimental points on the trendlines on traditional and membrane catalysts indicates the identity of chemical transformations on both catalysts. However, the quantitative ratios of the resulting products on MC and TC differ in small methane conversions, which indicates probable changes in the mechanisms of reagent transport or differences in the mechanism of the DRM process as a whole.

As can be seen in Figure 5, the concentration data in the membrane catalyst fit well with the trendlines for the corresponding components. The data for the traditional (powder) catalyst partly deviate from these lines, thereby indicating possible differences in the DRM process in the reactors under consideration.

These differences are most clearly manifested in the reactions of formation and consumption of hydrogen and water vapor (Figure 5 and Figure 6). On a membrane catalyst, the experimental points of H_2_ and H_2_O concentrations follow the trendlines for the consumption of CH_4_, which are involved in reactions III and IV. On a traditional catalyst, some of the points deviate from this trend dependence for MC. Similar deviations are observed in the dependences characterizing the formation of CO and H_2_. If during the formation of CO only some of the points deviate from the trendline for the membrane catalyst, then for hydrogen all the obtained values are located below the values for the membrane catalyst. That is, much less hydrogen is generated on a traditional catalyst than on a membrane catalyst. In principle, reactions II and IV are involved in the formation of hydrogen, reactions III, IV, and V are involved in the formation of CO (or reaction IV does not occur at all), and carbon interacts with CO_2_, forming carbon monoxide according to reaction V.

The dependence in Figure 6 depicts the change in the concentration of water vapor from the amount of the methane consumed. In the process of DRM, water vapor is an intermediate product and is formed in the reaction (III), both on traditional and membrane catalysts. The position and shape of the trendline for water on a membrane catalyst are determined by the results obtained in a wider range of flow rates of the reagent mixture (30–320 cm^3^/min) and the temperature range (820–890 °C).

For a traditional catalyst, the range of methane consumption is narrower, but the trendline for water vapor also has an extreme shape. As the methane conversion increases, the concentration of water vapor on both catalysts passes through a maximum before beginning to decrease, tending to zero as the methane conversion approaches 100% (Figure 6). For a traditional catalyst, the maximum value of the concentration of water vapor is observed at the initial portion of the trendline and decreases, approaching the trendline for the membrane catalyst. Since the phase composition of both catalysts is the same (Table 1), and the conditions of their study are similar, it should be noted that the final composition of the products of the DRM on the membrane and traditional catalysts is determined by the competing reactions (IV and V). On a membrane catalyst, hydrogen is formed in a more quantitative manner, and, therefore, in this case, reaction IV is more likely. On a traditional catalyst, reaction V is more likely, which is confirmed by studies of other authors [18,19,20,21]. Therefore, the reaction mixture contains water vapors in higher concentrations than on the membrane catalyst. Or, in other words, reaction IV precedes reaction V. It can only happen if water vapor is faster delivered to the catalyst surface than ${\mathrm{CO}}_{2}$. That is, if there is a change in the mechanism of transport of ${\mathrm{CO}}_{2}$ and $H_{2}O$ to the active centers of the catalyst occupied by carbon deposits.

### 3.5. Transport Characteristics of the Membrane Catalyst

Considering that a significant change in rate constant of reaction II was observed precisely on the membrane catalyst, then, in this part of the work, the transport characteristics of the reagents were determined at different temperatures, both in the presence of a chemical reaction and in its absence.

Powdered molybdenum carbide consists of particles with low porosity (Table 1), which is necessary for the experimental determination of the reaction rate constant in the kinetic region. The transport of reagents of such particles will proceed in accordance with Fick’s law.

As can be seen from the results presented in Table 3, the effective diffusion coefficients of CH_4_ and CO_2_ in the pore space of the membrane catalyst correspond to Knudsen diffusion. First, the effective diffusion coefficients established for different mixtures are practically the same, and the presence of the second component in the mixtures does not affect the diffusion of methane. Second, the ratio of the effective diffusion coefficients of carbon dioxide and methane turned out to be inversely proportional to the square root of the ratio of the molar masses of dioxide and methane. This experiment was carried out at an elevated temperature of 200 °C in order to reduce the surface diffusion factor while determining this characteristic in the pore structure of the membrane catalyst.

It is also seen from the data in Table 4 that at the temperatures of dry reforming of methane, the Knudsen number is greater than 10. This means that the transport of reagents in the channels of the pore structure of the membrane catalyst at 850 °C will be determined by the Knudsen diffusion law. In this case, the molecules of all components of the reaction medium, when moving in the pores of the membrane catalyst, will collide, mainly, with the pore walls. The absence of intermolecular interaction in the pore space of the membrane catalyst supposes in them the constant participation in reaction (III), which is homogeneous and is characterized by collisions of molecules with each other.

In accordance with Knudsen’s law, the flow rate of any component of the reaction mixture should decrease in proportion to T^−0.5^. However, from the data in Table 5, it can be seen that, under DRM conditions, the methane flow rate in the porous structure of the membrane catalyst, even without taking into account increase in gas viscosity with increasing temperature (850 °C), exceeds the methane flux at 200 °C established by the diaphragm method by an order of magnitude. The only reason for the significant difference in the methane flux densities in these two experiments is the temperature difference that occurs in a reactor with a membrane catalyst under the conditions of DRM.

The presence of a large endothermic effect in dry reforming of methane is the main reason for the appearance of a temperature difference in the tube side and shell side of the membrane reactor and a temperature gradient in the channels of the pore structure of the membrane catalyst.

Figure 7 shows the dependence of temperature differences on the amount of reacted methane at temperatures of 820–890 °C.

These dependencies of temperature are due to a summation of thermal effects of reactions of DRM. It can be assumed that the temperature gradient on the surface of the channels in the pore structure of the membrane catalyst and the law of its change are related to the rate of heterogeneous chemical reactions II, IV, and/or V occurring in the layer of the catalytic substance distributed on the membrane catalyst.

As shown in Ref. [9], the occurrence of such a temperature gradient under Knudsen diffusion in the channels of any pore structure is accompanied by an increase in the flow of matter as a result of a phenomenon called thermal creep, thermal transpiration, or thermal slip. This phenomenon is characterized by the transfer of gas molecules in the near-wall layer of micro- and nanochannels in the direction from a region of low temperatures and pressures to that with higher values of these parameters. The gas in the near-wall layer is in a rarefied state; its molecules mainly collide with the surface of the channels in the pore structure and practically do not interact with each other.

If we consider the cross-section of the reactor near the membrane catalyst, then it is easy to find the structural similarity of the membrane reactor with the Knudsen compressor. This analogy is shown in Figure 8.

Both devices have a porous diaphragm dividing the working volume into two parts—“cold” and “hot”. The working volumes of the devices are heated and cooled from different sides of the diaphragms. This leads to the appearance of a temperature gradient on the channel walls of the porous partition. In the compressor, this happens due to the presence of a cooler in the design of this device, and in the membrane reactor, due to the endothermic effects of chemical reactions. The principle of operation of the Knudsen compressor (Figure 7) has been known for a long time and is widely used in various microelectromechanical devices [22,23,24]. The temperature gradient in such a device is created artificially, for example, using a heater (Figure 7, item 1) and a cooler (Figure 7, item 2) due to the difference (Q1–Q2).

The temperature gradient on the surface of the channel walls generates flows of gas molecules in the near-wall layer of the channels (red arrows) from the “cold” volume to the “hot” one. As a result of the transfer of molecules in the near-surface layer of gas (thermal slip) in the channels of the porous partition in the cooled volume, differential pressure (P_hot_-P_cold_) is formed. This differential pressure causes the flow of gases in the opposite direction (viscous flow), from the heated volume (shell side of the reactor) already along the axis of the channel (blue arrows) to tube side. The thermal slip flow in the Knudsen compressor is the sum of the flows, which in the heated volume is divided into the viscous flow returning to the pore channels and the flow leaving the working volume of the compressor. At steady-state temperatures (under stationary conditions), a certain ratio is established between the near-wall gas flow and the flow returning to the cooled volume, which makes it possible to consider these flows in pore channels as a circulating loop. 

In the compressor, the temperature gradient on the surface wall of the channels is created by artificial heating and cooling of the chamber (Q1–Q2), and in the membrane reactor, by the endothermic effect of the reactions (∑*F_i_* ΔH_*reaction*_) occurring on the catalyst deposited on the porous support. It was shown in [25] that the transpiration phenomenon on a membrane catalyst can also be initiated by artificial creation of a temperature gradient in the channels of the pore structure. In the same place, the values of transpiration air flow through the channels of the pore structure of the membrane catalyst and the pressure drop arising across it were estimated.

Analysis of gas transport in a membrane catalyst reactor reveals that there are two independent driving forces acting. One, the pressure difference at the inlet and outlet moves the gas medium through the shell side of reaction space in the reactor, and the second, the temperature gradient on the walls of channel that moves the reaction medium through the pore structure of the membrane catalyst. As shown above (Figure 4 and Table 1), the acceleration of gas transport is accompanied by a significant decrease in the contact time of the gas mixture with the surface walls of the channels of the catalytic layer; on the other hand, a significant increase in the rate constant of reaction II is due to the multiple instances of contact of this mixture with the catalytic surface. The ratio of the residence times in the shell side volume of the reactor and the contact time with the surface of the catalytic bed can be considered as a measure of the DRM intensification.
(13)$$n=\frac{\tau_{res}}{\tau_{cont}}=\frac{\frac{V_{R}}{Q_{{\mathrm{CH}}_{4}}^{0}}}{\frac{V_{P}}{Q_{{\mathrm{CH}}_{4}}^{r}}}$$

n—coefficient of intensification of mass transfer in the pores of the membrane catalyst; $\tau_{res}$—residence time in the reaction volume; $\tau_{cont}$—contact time for methane molecules in the porous structure; $V_{R}$, $V_{P}$—the working volume of the reactor and the pore volume of the catalytic bed, respectively, m^3^; $Q_{{\mathrm{CH}}_{4}}^{0}$—flow rate of methane at the reactor inlet, m^3^/s; $Q_{{\mathrm{CH}}_{4}}^{r}$—flow rate of methane that participates in the reaction (from the material balance of the process).

Methane conversion was not chosen by chance to assess the intensification of the DRM process. It participates only in reaction II, which is recognized by many researchers as a stage that limits this process as a whole. Moreover, this reaction can only take place on the surface of the catalytic layer of the membrane.

## 4. Conclusions

The mass transfer in the micro- and nanochannels of the pore structure of various materials is of decisive importance for many technological processes. Mass transfer in porous materials for many chemical processes often becomes a factor that determines their economic efficiency. Membrane technologies are a source of examples of both positive and negative effects of mass transfer on economic performance. These examples include the experimental results presented in this work. The effectiveness of using membranes in catalysis from the very beginning did not raise doubts. The main way to achieve positive results was considered through the possibility of changing the thermodynamic characteristics of reversible chemical reactions, in particular, the equilibrium degree of conversion. However, for the majority of heterogeneous catalytic processes, the effect of using membrane appeared to be uneconomical. Positive examples of processes on membrane catalysts (catalytic membranes) can be found in the review by T. Westermann, T. Melin [3], but they are not because of changing of thermodynamic reactions.

Until recently, dry reforming of methane could be attributed to negative examples. This reaction is characterized not only by thermodynamic limitations due to catalyst deactivation at high temperatures, but also by kinetic limitations. The use of membrane catalysts in reactions accompanied by a noticeable thermal effect makes it possible to significantly accelerate the chemical process by intensifying mass transfer stages of the catalytic act.

The results obtained in this work not only demonstrate the effect of the non-isothermality of the catalytic surface on mass transfer, but also make it possible to explain the intensification of the catalytic process in the presence of a membrane catalyst using the modern theory of Knudsen non-isothermal diffusion. The mode of diffusion transport in a reactor with a membrane catalyst investigated in this work makes it possible to significantly reduce the hydraulic resistance, providing a high efficiency of the membrane catalyst used. Moreover, the DRM intensity in such a reactor turns out to be an order of magnitude higher than the intensity in a reactor with a conventional catalyst bed of the same chemical and phase compositions.

The phenomenon of thermal slip (thermal transpiration), which accelerates the transfer of gases into the catalyst pores and is accompanied by the circulation of the reaction medium in them, causes a change in the route in the multistage DRM process. In particular, due to the circulation of the gaseous medium caused by the temperature gradient in the channels of the pore structure of the membrane catalyst, the reaction of carbon deposits with water vapor becomes more preferable than with carbon dioxide. A change in the transport mechanism in the pore structure of a membrane catalyst leads not only to an intensification of DRM as a whole, but also to a change in the mechanism in comparison with a traditional catalyst. Due to the circulation of the gaseous medium on the membrane catalyst, the reaction of carbon deposits with water vapor becomes more preferable, and the carbon dioxide reaction with hydrogen becomes a source of water vapor for the conversion of methane.

In the hypothesis set forth in [7], which is based on the peculiarities of gas transport in non-isothermal conditions in the channels of the pore structure, it is noted that the chemical reaction under these conditions is coupled with the transport of gases. The theory of the rarefied gases transport under the conditions of a temperature gradient in micro- and nanochannels substantiates the possibility of the mass transfer even in the absence of a pressure difference on both sides of a porous diaphragm. The results obtained in this work confirm this fact. It can be assumed that the use of reactors with membrane catalysts will make it possible to create effective devices for producing synthesis gas from natural and associated petroleum gases.

The experiment presented in the work and the analysis of the results obtained, taking into account the peculiarities of transport in non-isothermal conditions, allow us to not only explain the intensification of heterogeneous reactions on a membrane catalyst, but also to more meaningfully approach the design of catalytic processes in reactors with a membrane catalyst.

## Figures and Tables

**Figure 1 membranes-11-00497-f001:**
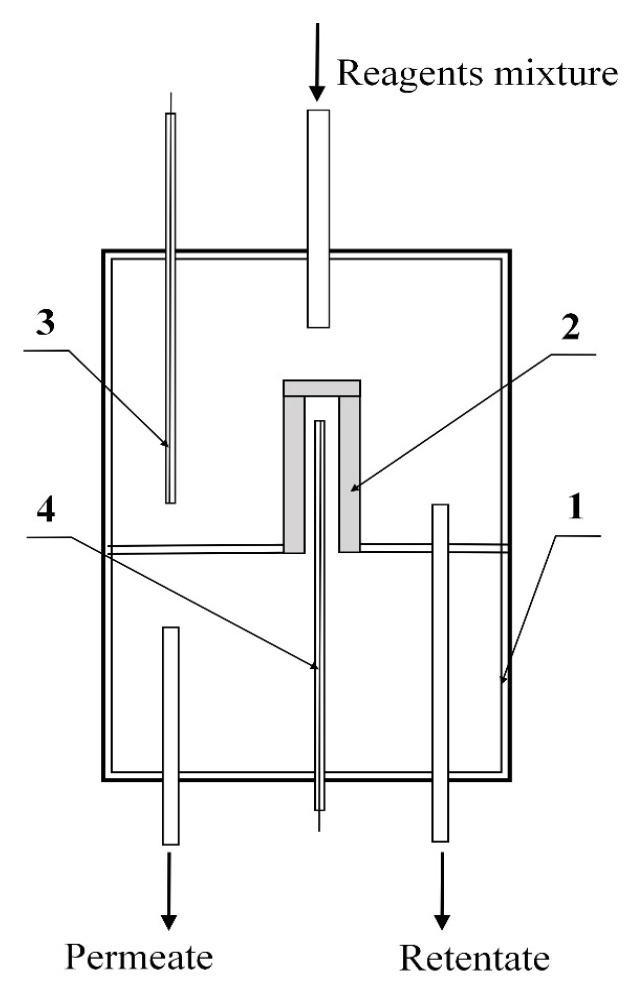
Schematic diagram of a reactor with a membrane catalyst: 1—membrane reactor, 2—membrane catalyst, 3, 4—thermocouples that control temperatures outside (in the shell side of reactor) and inside (tube side) of membrane catalyst.

**Figure 2 membranes-11-00497-f002:**
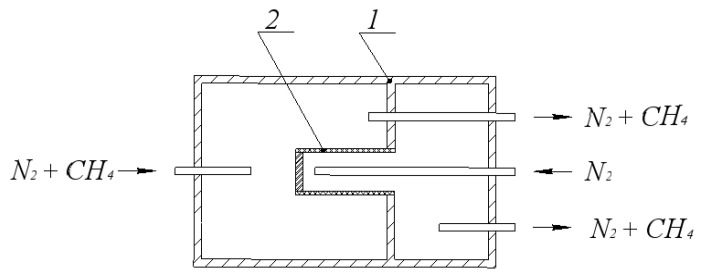
Diagram of a membrane reactor for determining the effective diffusion coefficient of methane in a mixture with nitrogen: 1—membrane reactor, 2—diaphragm (membrane catalyst).

**Figure 3 membranes-11-00497-f003:**
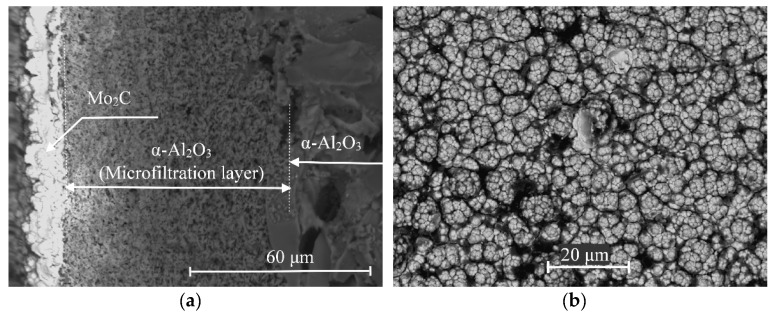
SEM images of the cross-section (**a**) and the outer surface (**b**) of the membrane catalyst.

**Figure 4 membranes-11-00497-f004:**
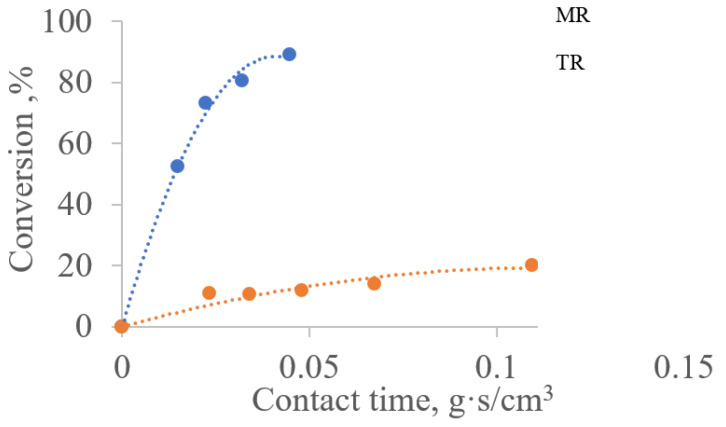
Dependence of the degree of methane conversion on the contact time for traditional and membrane reactors at a temperature of 850 °C:TR—reactor with a traditional catalyst; MR—reactor with membrane catalyst in diffusion transport mode.

**Figure 5 membranes-11-00497-f005:**
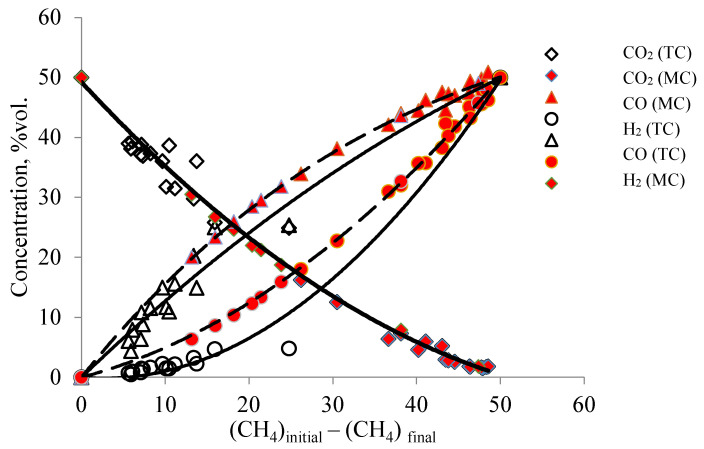
Dependence of the change in the concentration of the products of the DRM on the amount of methane that participated in reaction (II) on membrane and traditional catalysts in the temperature range (conditions: T (MC) from 820 °C to 890 °C, volumetric flow rate of the mixture 30–320 cm^3^/min; T (TC) from 830 to 900 °C, volumetric flow rate of the reagent mixture 50–150 cm^3^/min.

**Figure 6 membranes-11-00497-f006:**
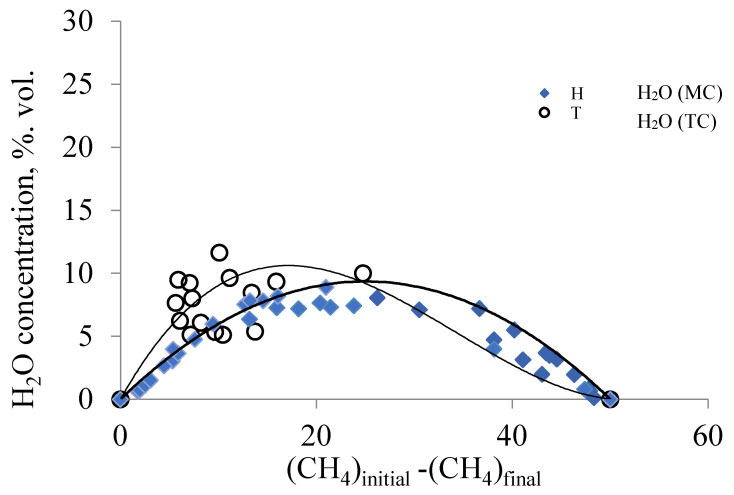
Change in the concentration of water vapor from the amount of the methane consumed on the traditional (TC) and membrane catalysts (MC). (Conditions: T (MC) from 820 °C to 890 °C, flow rate of the mixture 30–320 cm^3^/min; T (TC) from 850 °C to 900 °C, flow rate of the reagent mixture 50–150 cm^3^/min).

**Figure 7 membranes-11-00497-f007:**
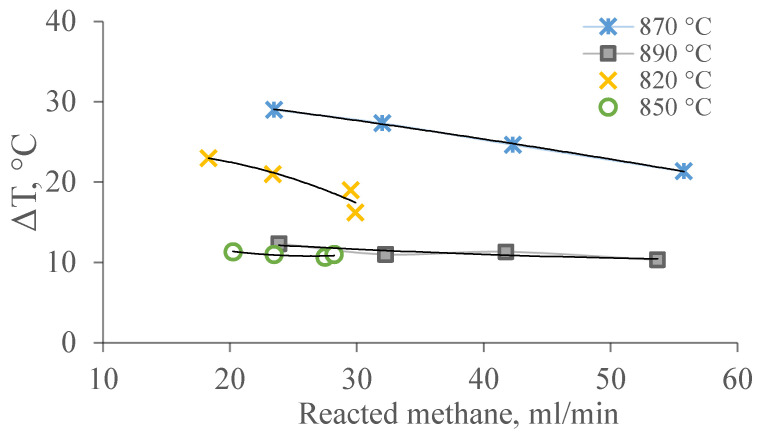
Dependence of the temperature gradient on the amount of methane consumed under the conditions of the DRM at temperatures of 820–890 °C.

**Figure 8 membranes-11-00497-f008:**
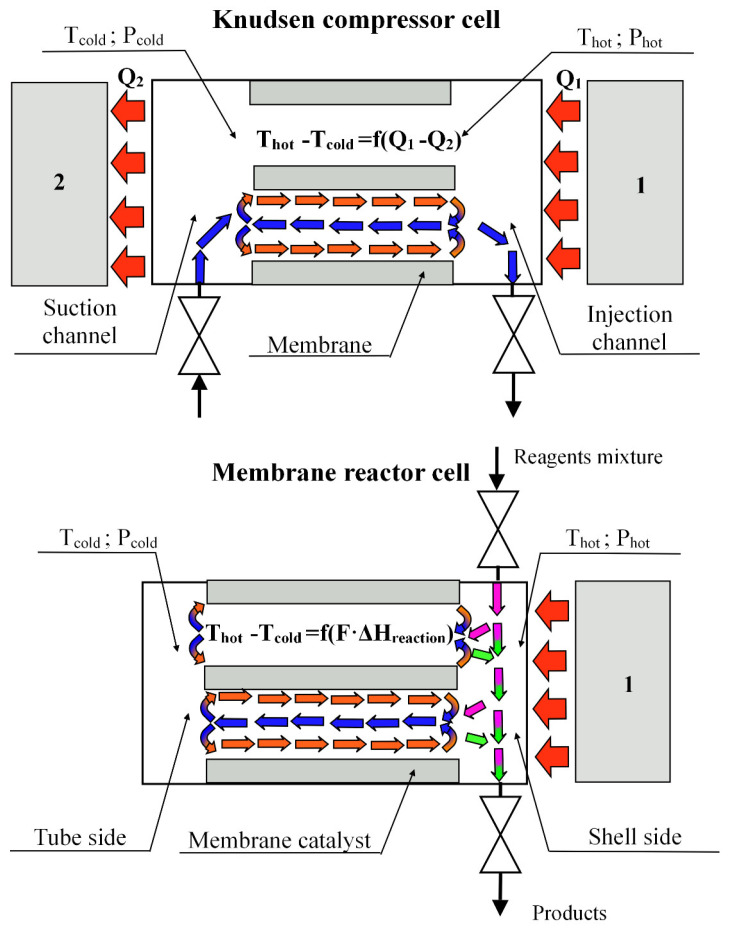
Constructive analogy and flow diagram in a Knudsen compressor and a reactor with a membrane catalyst (cross-section near diaphragm): 1—heater, 2—cooler. $F_{i}$—mole flow of component, which takes part in the endothermic reaction; $\Delta H_{reaction}$—heat effect of reaction. Mass flows in the shell side: pink arrows—reagents mixture, green arrows—products of DRM; circulation loop: orange arrows—thermal slip flow; blue arrows—viscous flow. Thermal flow—red arrows.

**Table 1 membranes-11-00497-t001:** Reactor loading, phase composition, and characteristics of the pore structure of traditional and membrane catalysts.

Catalyst Type	Loading, g	Phase Composition	Surface Area, m^2^/g	Pore Volume cm^3^/g
Traditional	0.262	β-Mo_2_C, η-MoC	3.2	0.010
Membrane	-	α-Al_2_O_3_, β-Mo_2_C, η-MoC	0.8	0.053
Catalytic layer MC	0.154	β-Mo_2_C, η-MoC	8.2	0.074

**Table 2 membranes-11-00497-t002:** Comparison of parameters of DRM in conventional and membrane reactors.

Reactor Type	Catalyst Type	$E_{app}$, kJ/mol (CH_4_)	Specific Rate Constant, cm^3^/(g·s)
membrane reactor-contactor with diffusion transport			
	membrane	115	48

conventional reactor	powder	264	1.2

**Table 3 membranes-11-00497-t003:** Effective diffusion coefficients of methane and carbon dioxide, established experimentally under isothermal conditions (at a temperature of 200 °C by the diaphragm method).

Mixture Composition	Diffusing Component	*D_ef_* 10^6^, m^2^/s	$D_{ef}^{{\mathrm{CO}}_{2}}$/$D_{ef}^{{\mathrm{CH}}_{4}};\;$	$\sqrt[{2}]{M\frac{\left({\mathrm{CH}}_{4}\right)}{M\left({\mathrm{CO}}_{2}\right)}}$
N_2_:CH_4_ = 1:1	CH_4_	5.33	1.63	1.61
CO_2_:CH_4_ = 1:1	CH_4_	5.42	1.66	1.61
N_2_:CO_2_ = 1:1	CO_2_	3.26	-	-

**Table 4 membranes-11-00497-t004:** The value of the Knudsen number at the temperature of 850 °C for the components of the reaction mixture of DRM.

Component	Molecule Free Path (λ) 10^9^, m	Knudsen Number Kn=λ/d *
CO_2_	320	20.9
CH_4_	359	23.5
CO	394	25.8
H_2_O	460	30.0
H_2_	754	49.3

* d—mean pore diameter, m.

**Table 5 membranes-11-00497-t005:** Comparison of the results of the kinetic experiment and the experiment to determine the effective diffusion coefficient of methane *.

Temperature, °C	Kinetic Experiment $J_{kin.}*10^{3}$, $\frac{\mathrm{mol}}{s*m}$	Reuter Method * $J_{dif}*10^{3}$, $\frac{\mathrm{mol}}{s*m}$
820	2.20	0.37
850	2.30	0.37
870	2.53	0.37
890	2.47	0.37

* Note: the diffusion flux in the structure of the membrane catalyst $J_{dif}$ was set in the experiment at a temperature of 200 °C and is presented in the table without taking into account the temperature coefficient.

## Data Availability

The data presented in this study are available on request from the corresponding author.

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
