# Peer review of "Transport Reagents through the Pore Structure of a Membrane Catalyst under Isothermal and Non-Isothermal Conditions"

_membranes, 2021, doi:10.3390/membranes11070497_

Round 1

Reviewer 1 Report

This manuscript has serious flaws. The storyline was messy. The figures presented were unprofessional. There were spelling errors throughout the entire manuscript. Abbreviations were used without first spelling out what they meant. Parts of the manuscript were missing. Result discussion was all over the place.

This manuscript is not English-ready for publication. It needs to be thoroughly revised by a native English speaker for proper English language, spelling, grammar, punctuation, and overall style.

What does DRM in the abstract stand for? Please do not abbreviate words without first letting your readers know what the abbreviation stands for.

Incorrect spelling is a major issue. Line 9 of the first paragraph of the Introduction Section, ‘construction’ not ‘konstruction’. Please check other parts of the manuscript. ‘separately’ not ‘separatly’. ‘realised’ not ‘realysed’.

The entire Section 2.1 is missing.

Figure 3, please add scale bars. Label what each layer represents in the cross-section image.

Figures 4, 5, and 6, what do the symbols represent? Please include a legend.

Author Response

This manuscript has serious flaws. The storyline was messy. The figures presented were unprofessional. There were spelling errors throughout the entire manuscript. Abbreviations were used without first spelling out what they meant. Parts of the manuscript were missing. Result discussion was all over the place.

The manuscript and figures were improved, section 2.1 was added.

This manuscript is not English-ready for publication. It needs to be thoroughly revised by a native English speaker for proper English language, spelling, grammar, punctuation, and overall style.

The manuscript was improved by a native English speaker.

What does DRM in the abstract stand for? Please do not abbreviate words without first letting your readers know what the abbreviation stands for.

Improved.

Incorrect spelling is a major issue. Line 9 of the first paragraph of the Introduction Section, ‘construction’ not ‘konstruction’. Please check other parts of the manuscript. ‘separately’ not ‘separatly’. ‘realised’ not ‘realysed’.

Improved

The entire Section 2.1 is missing.

Section 2,1 was added.

Figure 3, please add scale bars. Label what each layer represents in the cross-section image.

Corrected.

Figures 4, 5, and 6, what do the symbols represent? Please include a legend.

Legends were included. 

Reviewer 2 Report

This paper developed a membrane reactor for dry reforming of methane (DRM) and tested the reaction under both isothermal and non-isothermal conditions. The kinetics of the reaction and diffusion coefficients were measured. The paper can be accepted for publication if the authors can address the following comments.

  1. The activation energy obtained by the membrane reactor (115 KJ/mol) is significantly lower than that obtained by traditional reaction (264 KJ/mol). The reviewer thinks that it is very unlikely unless the chemistry of the DRM changed. Could the authors explain this observation (how did the mechanism change)? Is there any literature that found similar results?
  2. The reviewers found that it was quite challenging to read some figures in the manuscript such as Figures 4-6 due to the missing figure legends. The authors should provide sufficient figure legends to make the plots and curves in these figures readable.
  3. The rete constants were determined in Table 2. What are the reaction orders of CH4 and CO2 for DRM reaction tested in the membrane reactor?
  4. In the abstract of the paper, the authors should add specific numbers to support the increased rates of transport and the intensified process. Additionally, the full name of DRM, i.e., dry reforming of methane should be added to the abstract to avoid confusion.

Author Response

The authors are grateful to the Reviewer for such a careful reading of the manuscript and comments. The corrections will make the article better.

The activation energy obtained by the membrane reactor (115 KJ/mol) is significantly lower than that obtained by traditional reaction (264 KJ/mol). The reviewer thinks that it is very unlikely unless the chemistry of the DRM changed. Could the authors explain this observation (how did the mechanism change)? Is there any literature that found similar results?

The decrease in the value of the apparent activation energy in our experiment on a membrane catalyst is quite consistent with the conclusions that can be found in the monograph [I. Chorkendorff, J.W. Niemantsverdriet. “Concept of Modern catalysis and kinetics” / (2007) Wiley-VCH Verlag & Co.KGAA Weinheim. It indicates that the catalytic process on a membrane catalyst occurs under transport constraints, as opposed to conditions on a traditional catalyst. It indicates that the catalytic process on a membrane catalyst occurs under conditions of transport restrictions, in contrast to the conditions on a traditional catalyst. In order to ensure the absence of diffusion restrictions on a traditional catalyst (on Mo2C powder), it was mixed with crushed quartz glass. An increase in the reaction rate constant (I) in our experiment on a membrane catalyst indicates that there is an increase in the catalyst efficiency coefficient η, which occurs as a result of an increase in the diffusion coefficient D_eff (in accordance with the formula)

However, in the case of a membrane catalyst, the driving force of diffusion in the pores is the temperature gradient on the channel walls in its pore structure, and not the concentration difference. Therefore, this equation can only be used for a qualitative analysis of our experimental results

The reviewers found that it was quite challenging to read some figures in the manuscript such as Figures 4-6 due to the missing figure legends. The authors should provide sufficient figure legends to make the plots and curves in these figures readable.

The legends were added. The detailed information for the Figures 4-6 is presented in descriptions.

The rаte constants were determined in Table 2. What are the reaction orders of CH4 and CO2 for DRM reaction tested in the membrane reactor?

The manuscript contains only the constants of the initial speed (1). This reaction by the general scientist group, works in this area, is considered the limiting stage of the DRM process, i.e. determining the rate of the entire DRM process. The CH4 order is the first, it has been established experimentally and is in good agreement with the literature data. The order of the reaction with respect to CO2 was not determined.

In the abstract of the paper, the authors should add specific numbers to support the increased rates of transport and the intensified process. Additionally, the full name of DRM, i.e., dry reforming of methane should be added to the abstract to avoid confusion.

Abstract was improved.

Reviewer 3 Report

The work under review reports on an experimental comparison of methane transport in the pore structure of a membrane catalyst in DRM process. In my opinion the works is written very well and can be published after very minor corrections, including English spell check (some errors in spelling are listed below, anywhow I adivce to revise the spelling in the entire manuscript).

Minor comments:

  1. Abstract: the acronym DRM must be explained when used for the first tim.
  2. P2, second paragraph: should be "studied" instead of "studed"
  3. P3: third paraghrap: should be "premixed"
  4. P4: after eq. 1, m in km should be in lower case:
  5. P5: paragraph 6 - please define acronym CVD
  6. P9: paragraph 1: should be "activation" not "activatrd"
  7. P9: I guess that both the rate constants and activation energies described in the paragraph 1 and reported in Table 2 are the apparent values (as they account also for transporth phenomena) - this should be clearly said.
  8. P12: please give a definition of the Knuden number
  9. P12: what is the sense of determination of effective diffusivity at so low temperature?

Author Response

The work under review reports on an experimental comparison of methane transport in the pore structure of a membrane catalyst in DRM process. In my opinion the works is written very well and can be published after very minor corrections, including English spell check (some errors in spelling are listed below, anyhow I advice to revise the spelling in the entire manuscript).

The authors are grateful to the Reviewer for such a careful reading of the manuscript and comments. The corrections will make the article better.

Minor comments:

Abstract: the acronym DRM must be explained when used for the first time.

Improved

P2, second paragraph: should be "studied" instead of "studed"

Improved

P3: third paraghrap: should be "premixed"

Improved

P4: after eq. 1, m in km should be in lower case:

Improved

P5: paragraph 6 - please define acronym CVD

Improved

P9: paragraph 1: should be "activation" not "activatrd"

Improved

P9: I guess that both the rate constants and activation energies described in the paragraph 1 and reported in Table 2 are the apparent values (as they account also for transport phenomena) - this should be clearly said.

Corrected

P12: please give a definition of the Knudsen number

Corrected

P12: what is the sense of determination of effective diffusivity at so low temperature?

The determination of the effective diffusion coefficient under isothermal conditions was carried out at 200°C due to thermal dissociation of methane at membrane catalyst at DRM temperatures.

Round 2

Reviewer 1 Report

Figures 4, 5, 6, and 7 need major improvement. Legends are confusing.

There are incomplete and incoherent sentences in the manuscript.

There are still spelling errors.